# Healthcare-Acquired Infection Surveillance in Neurosurgery Patients, Incidence and Microbiology, Five Years of Experience in Two Polish Units

**DOI:** 10.3390/ijerph19127544

**Published:** 2022-06-20

**Authors:** Elżbieta Rafa, Małgorzata Kołpa, Marta Zofia Wałaszek, Adam Domański, Michał Jan Wałaszek, Anna Różańska, Jadwiga Wójkowska-Mach

**Affiliations:** 1State Higher Vocational School in Nowy Sącz, 33-300 Kraków, Poland; raela@vp.pl; 2Academy of Science in Tarnow and St. Luke’s Provincial Hospital in Tarnów, 33-100 Tarnów, Poland; malgorzatakolpa@interia.pl (M.K.); mz.walaszek@gmail.com (M.Z.W.); 3Department of Distributed Systems and IT Equipment, Faculty of Automatic Control, Electronics and Computer Science, Silesian University of Technology, 44-100 Gliwice, Poland; adam.domanski@polsl.pl; 4Polish Society of Hospital Infections, 31-121 Krakow, Poland; michalj.walaszek@gmail.com; 5Department of Microbiology, Jagiellonian University, 31-121 Kraków, Poland; mbmach@cyf-kr.edu.pl

**Keywords:** healthcare-associated infections, neurosurgery, benchmarking, infection prevention and control

## Abstract

Introduction: Patients in neurosurgical units are particularly susceptible to healthcare-associated infections (HAI) due to invasive interventions in the central nervous system. Materials and methods: The study was conducted between 2014 and 2019 in neurosurgery units in Poland. The aim of the study was to investigate the epidemiology and microbiology of HAIs and to assess the effectiveness of surveillance conducted in two hospital units. Both hospitals ran (since 2012) the unified prospective system, based on continuous surveillance of HAIs designed and recommended by the European Centre for Disease Prevention and Control (protocol version 4.3) in the Healthcare-Associated Infections Surveillance Network (HAI-Net). In study hospitals, HAIs were detected by the Infection Prevention Control Nurse (IPCN). The surveillance of healthcare infections in hospital A was based mainly on analysis of microbiological reports and telephone communication between the epidemiological nurse and the neurosurgery unit. HAI monitoring in hospital B was an outcome of daily personal communication between the infection prevention and control nurse and patients in the neurosurgery unit (HAI detection at the bedside) and assessment of their health status based on clinical symptoms presented by the patient, epidemiological definitions, microbiological and other diagnostic tests (e.g., imaging studies). In hospital A, HAI monitoring did not involve personal communication with the unit but was rather based on remote analysis of medical documentation found in the hospital database. Results: A total of 12,117 patients were hospitalized. There were 373 HAIs diagnosed, the general incidence rate was 3.1%. In hospital A, the incidence rate was 2.3%, and in hospital B: 4.8%. HAI types detected: pneumonia (PN) (*n* = 112, 0.9%), (urinary tract infection (UTI) (*n* = 108, 0.9%), surgical site infection (SSI) (*n* = 96, 0.8%), bloodstream infection (BSI) (*n* = 57, 0.5%), gastrointestinal system infection (GI) (*n* = 13, 0.1%), skin and soft tissue (SST) (*n* = 9, 0.1%). HAI with invasive devices: 44 ventilator-associated pneumonia (VAP) cases (45.9/1000 pds with ventilator); catheter-associated urinary tract infection (CA-UTI): 105 cases (2.7/1000 pds with catheter); central venous catheter (CVC-BSI): 18 cases (1.9/1000 pds with CVC). The greatest differences between studied units were in the incidence rate of PN (*p* < 0.001), UTI (*p* < 0.001), and SSI (*p* < 0.05). Conclusions: The way HAIs are diagnosed and qualified and the style of work of the infection control team may have a direct impact on the unit epidemiology with the application of epidemiological coefficients. Prospective surveillance run by the infection prevention and control nurse in hospital B could have been associated with better detection of infections expressed in morbidity, especially PN and UTI, and a lower risk of VAP. In hospital A, the lower incidence might have resulted from an inability to detect a UTI or BSI and less supervision of VAP. The present results require further profound research in this respect.

## 1. Introduction

Patients in neurosurgical units are at risk of healthcare-associated infections (HAIs), mainly surgical site infections (SSI) and other types of treatment-associated infections [1]. In neurosurgery units, HAIs are a serious problem, especially in critically ill patients [2]. A high risk of HAIs is associated with medical interventions in the central and peripheral nervous system, and brain injury is a particularly strong predictor of HAIs [3], generating problems such as coma, hyperglycemia, and hypothermia. Therefore, the care of the neurocritical patient aims at avoiding factors that cause secondary brain damage, which is one of the elements of prevention and control of HAIs in this patient group [4]. Other risk factors for HAIs are the use of invasive devices such as central venous catheters (CVC), urinary catheters (UC) and mechanical ventilation (MV) [5,6].

Data on HAI in neurosurgery are relatively scarce. The available studies estimate the incidence at the level of 2% to 14% and include patients hospitalized in neurosurgical departments with intensive care [6,7,8]. The incidence of HAI is much higher in neurosurgical intensive care units (ICU)-an Italian study [9] shows a result of 22% and 29% was recorded by Indian researchers [10].

Common clinical forms of HAIs among this patient population include surgical site infection (SSI), pneumonia (PN), bloodstream infection (BSI) and urinary tract infection (UTI) [8,11]. The present study demonstrates general incidence rates of SSI without an in-depth analysis of this form of infection because SSIs were not the main subject of this report. The essence of the study concerned HAI resulting from the use of invasive devices, which were risk factors of VAP, CA-UTI and CVC-BSI in the units under study. The Polish literature includes studies on SSI in neurosurgery [5,7,11], however, there is a lack of papers describing the problem of HAI with the utilization of invasive devices, such as VAP, CA-UTI and CVC-BSI.

## 2. Materials and Methods

Both hospitals are located in large cities that are not regional capitals and are among the largest hospitals in their regions. They are not related administratively, and both are teaching hospitals that receive elective and emergency patients. The structure of the analyzed units included intensive surveillance beds, which are not as well equipped as the beds in basic (general) ICU. In the event of deterioration of their health, neurosurgical patients from these rooms are transferred to the general ICU ward. Unit A had 35 beds (including 5 intensive surveillance beds) and Unit B had 28 beds (including 3 intensive surveillance beds). Both hospitals had hospital infection teams with years of experience in infection prevention and control (Table 1). The surveillance of healthcare infections in hospital A was based mainly on analysis of microbiological reports, databases on patients and telephone communication between the epidemiological nurse and the neurosurgery unit. The HAI monitoring in hospital B was an outcome of daily personal communication between the infection prevention and control nurse and patients in the neurosurgery unit (HAI detection at the bedside) and assessment of their health status based on clinical symptoms presented by the patient, epidemiological definitions, microbiological and other diagnostic tests (e.g., imaging studies). Within the structures of both hospitals, there are local hospital microbiological laboratories, which is very rare in Poland.

Both hospitals have participated in the unified prospective system, based on continuous surveillance of HAIs (since 2012), implemented in accordance with the protocol and definitions of the Healthcare-Associated Infections Surveillance Network (HAI-Net), European Centre for Disease Prevention and Control (protocol version 4.3), which concerns the ICU [12].

The epidemiological analysis of the data took into consideration the following indicators [13,14]:HAI incidence rate, calculated as: (N of HAI × 100)/N of operations;CA-HAI incidence density rate: N of CA-HAI*1000/N of catheter days (CVC or urinary catheter, respectively);VAP rate: N of VAP*1000/N of mechanical ventilation days;Invasive procedure utilization, urinary catheter or CVC or mechanical ventilation use rate: N of days with invasive procedure/N of patient–days.Utilization ratio (UR)–number of patient-days/number of patient-days with a given (invasive) procedure.The epidemiology of gastrointestinal tract infection, including *Clostridioides difficile* infection (GI-CDI) and skin and soft tissue infection (SST), were not analyzed due to the low number of cases detected, respectively, 13 and 9 cases.

For the microbiological diagnosis of different HAI cases, appropriate clinical material (blood, swabs, urine samples and others) was collected following doctor’s orders. Only the first isolate from each patient was selected for microbiological analysis, excluding subsequent cultures from the same patient and HAI case. The strains were identified using BD Phoenix NID cards of the automated Phoenix 100 Becton Dickinson Diagnostic System (Beckton Dickinson, Warsaw, Poland) according to the manufacturer’s instructions, without molecular techniques.

In the study period, years 2014–2019, data on SSI were collected according to HAI-Net ECDC (protocol, version 2.2). This surveillance provided data on the number of surgeries and the incidence of SSI. Both units conducted the following operations: laminectomy (LAM) *n* = 3092; spinal fusion (FUSN) *n* = 1131; craniotomy (CRAN) *n* = 1841; Ventricular Shunt Operations (VSHN) *n* = 169. Central nervous system infections–meningitis or ventriculitis (CNS-MEN) were detected and reported as organ surgical site infections SSI-O [15]. Data obtained from surveillance of SSI were not the objective of this study.

IBM SPSS (SPSS-Statistical Package for the Social Sciences, STATISTICS 24, Armonk, NY, USA) and Microsoft Excel (Microsoft Office 2016 Redmond, WA, USA) were used in the statistical analysis of the collected material.

Statistical analysis presents descriptive statistics for the characteristics of hospitals, units, patients, and types of HAI.

A descriptive analysis of the qualitative characteristics was performed by calculating the number and percentage for each value. To characterize the age and duration of the patients’ stay, the quartile distribution due to the skewed distribution was presented. The analysis of differences between the analyzed departments was performed using Pearson’s chi-square. Standard measures of Odds Ratio (OR) and significance level (*p*) were calculated for two groups of variables tested, each of which was sorted according to the absence or presence of a particular feature. The level of significance was *p* < 0.05.

The test power calculated a posteriori was: 1, α = 0.05 for the hypotheses H0: HAI incidence (Z) is the same in the studied departments ZA = ZB and H1: there is a difference in the incidence of HAI in the studied departments ZA ≠ ZB. The number of patients required to obtain the calculated power of 0.8 should be 938 patients in each group. The test power for VAP, BSI and UTI was 1 (α = 0.05). Test power and sample size were calculated in STATISTICA version 13.3.

The data used for the present analysis had been previously anonymized. The use of data was approved by the Bioethical Committee of the Jagiellonian University (No. KBET/122.6120.118.2016 of 25 May 2016). All the data entered into the electronic database and analyzed in this study were previously anonymized.

## 3. Results

In the study period, a total of 12,117 patients were hospitalized, including 8347 in hospital A and 3770 in hospital B. A total of 11,306 surgical procedures were performed, including 7799 in hospital A and 3507 in hospital B. The length of hospitalization was significantly shorter in hospital A (Me 22 days/IQR 10–40) than in hospital B (Me 32 days/IQR 22–52) (*p* < 0.001).

The patients treated in hospital A (Me 57/IQR 45–68) were on average 7 years younger than patients treated in hospital B (Me 64/IQR 42–76), *p* < 0.001. The utilization of invasive procedures such as central line catheters, urinary catheters and mechanical ventilation was similar in both hospitals (Table 1).

A total of 373 HAIs were diagnosed and the incidence rate was 3.1% in total, but in each hospital, the rates were different: in hospital A it was 2.3%, and in hospital B 4.8%, OR 2.1 (1.73–2.58) *p* < 0.001, the incidence density rates were 2.8 per 1000 pds and 5.2 per 1000 pds, respectively (*p* < 0.001).

The most common HAI was pneumonia (112 cases, incidence rate 0.9%), significantly higher in hospital B (OR1.9 95%CI 1.3–2.8 *p* < 0.001). In addition, for urinary tract infections, 108 cases, the incidence rate of 0.9%, was significantly more often diagnosed in hospital B (OR 8.9, 95%CI 5.7–14.8; *p* < 0.001). Similarly, the incidence rates of surgical site infections were different, in hospital A, they were much higher than in hospital B (OR1.4; 95%CI 0.8–2.5; *p* < 0.05). The epidemiology of bloodstream infections was the same, a total of 57 cases, and incidence rates were 0.5% (Table 2).

The catheter-associated urinary tract infection (CA-UTI) and central venous catheter-bloodstream infection (CVC-BSI) rates were found higher for hospital B, while in the case of VAP incidence rates, the opposite was observed. There were 82 CA-UTIs found in hospital B (incidence per 1000 UC pds 4.5) while only 21 cases of CA-UTIs were observed in hospital A (incidence per 1000 UC pds 1.0). There were 11 cases of CVC–BSIs in hospital B (incidence per 1000 CVC pds 2.8), and 7 cases in hospital A (incidence per 1000 CVC pds stated at 1.3). There were 9 VAP cases detected in hospital B (incidence per 1000 MV pds 23.7) and 35 in hospital A (incidence per 1000 MV pds stated at 60.3) (Table 3).

The microbial etiology of infections depended on both, their clinical form and the hospital. For BSIs only, *Gram-positive cocci* predominated in both hospitals. For PNs, *Enterobacteriaceae* predominated, but there were significant differences in the microbiology of PN in the two units, both in the proportion of microbiologically unconfirmed cases (38% in hospital A and 79% in hospital B) and in the proportion of *Acinetobacter baumannii*, which accounted for 21.7% in hospital A vs. 1.9% in hospital B. *Enterobacteriaceae* were predominant in unit B. For UTIs, *Enterobacteriaceae* were dominant in both hospitals, accounting for more than half of the cases, but 19 UTI cases (22%) were not microbiologically confirmed in hospital B. In addition, in the case of SSIs, there were differences in predominance, in hospital A *Staphylococcus aureus* was the most frequently isolated (20 cases, 26.0%) while in hospital B, it was *Klebsiella pneumoniae* (5 cases, 26.3) (Table 4).

## 4. Discussion

Our study showed that the incidence rate of HAIs was 3.1%, but in one hospital it was about two times higher than in the other, 2.3% in hospital A and 4.8% in hospital B. It seems that these differences can result directly from the organization of IPCNs’ work in both hospitals. The differences in the rates were mainly related to UTI and PNU-clinical forms of infections for which diagnosis depends to a large extent on close cooperation with the patient. An example may be a UTI where pain or burning when urinating, will be a significant clinical symptom that cannot be detected without a direct relationship with the patient or the ward staff. Similarly, in the case of PN, clinical symptoms such as coughing or changing the nature of the discharge cause the patient to be observed and, consequently, may lead to the diagnosis of an infection. Thus, the lack of IPCN visits to ward A was probably the reason for the loss of information about infections that may not have been recognized and qualified by ward personnel. The situation is often described and defined as the so-called passive surveillance, but only in the initial period of implementing infection control programs, i.e., in the 1980s and 1990s. Unfortunately, such a different approach to HAI monitoring despite the use of the same ECDC methodology, definitions and protocols, has a direct significant impact on incidence rates–the difference in incidence rates of UTIs was almost ninefold.

A higher incidence of HAI was observed in hospital B, which should not be associated with a high risk of infection, but rather with more effective detection of HAIs diagnosed also without microbiological confirmation-the high efficiency of the infection control team: frequent personal contact with unit staff, including detection of HAI at the bedside and assessment of patients’ health status based on clinical symptoms presented by them in conjunction with microbiological and other test results, including imaging. Thanks to the above, high detection rates of infections have been achieved, including with microbiologically unconfirmed cases. The differences detected should be the subject of further studies encompassing an in-depth analysis of the system of surveillance of HAI in the units under study.

Unfortunately, on the basis of the obtained results, it cannot be concluded that the effectiveness of HAI surveillance in the studied departments is the same, despite the adopted HAI detection methodology, which was the same in both units and compliant with ECDC. This hypothesis requires further research in order to be proved or ruled out. As shown in Table 4, it is only possible to show that 78% of PN in hospital B were not microbiologically confirmed and thus required a diagnosis by IPCN based on evidence other than microbiological examination. Similarly, in the case of UTIs, 22% of cases were diagnosed without microbiological confirmation.

Significant differences were also observed in VAP rates, 60 and 24 cases per 1000 MV pds– both unfortunately significantly higher than the literature data, e.g., 11 per 1000 MV-pds in neurosurgical ICUs in the USA between 1992 and 2004 [16], and in 2012 it decreased to 2 per 1000 MV pds [14]. In Italy, the incidence was 11 per 1000 MV-pds in neurosurgical ICUs [9]. It is difficult to explain the disproportionately high-density rates of VAP reported in our hospitals. The cited references are based on neurosurgical ICUs, which may suggest that VAP prevention in these units is standardized and strictly observed. In the hospitals studied, patients requiring intensive care, including those requiring mechanical ventilation, were hospitalized in the neurosurgery unit-in intensive care rooms with the medical team of the neurosurgery department. Thus, such a solution is not optimal for the safety of the neurosurgical patient, who particularly requires the implementation of procedures aimed at minimizing the risk of VAP.

In Poland, there have been few publications addressing HAI in neurosurgery. Wieder–Huszla et al. [7] report the incidence of HAIs at 2% in a single-center research study conducted in 2004–2008 in Poland. In another Polish single-center research study conducted by Wałaszek [5] between 2003 and 2012 on a group of 13,351 patients, the incidence was 4%. The results obtained in the two independently working neurosurgical units studied by the authors differ by more than twice: 2.3 vs. 4.8. Such results were obtained despite the fact that similar surveillance results would normally be expected: both centers have implemented uniform protocols for infection prevention and control and both have long experience in HAI surveillance.

The countries of Central and Eastern Europe, which include Poland, are in the first stage of reform and are still building surveillance systems and creating an infection control infrastructure with European surveillance programs. Weak commitment, resource scarcity and a lack of expertise were identified. It is common to undercut official infection control statistics [17].

Worrying results obtained in our study concern the high prevalence of *Klebsiella*
*pneumoniae* strains in the etiology of UTI (hospital A 19%, hospital B 21%). In the ECDC report published in 2012 [18], *Klebsiella pneumoniae* strains accounted for 6% of all isolates cultured from UTIs. These concerns seem to be confirmed by the results obtained in the 2016–2019 HAI survey in ICUs of southern Poland, where it was shown that *Klebsiella pneumoniae* and *Acinetobacter baumannii* were the most frequently isolated microorganisms [19].

The slightly less frequently identified *Acinetobacter baumannii*, which accounted for only 9% of the total number of cases but which is becoming one of the most dominant problems due to its potentially high drug resistance, seems particularly worrying [20]. Unfortunately, *Acinetobacter baumannii* occupies an important place in the etiology of HAIs in neurosurgery. An Indian study conducted in the years 1995–1996 showed that this pathogen was responsible for ¼ of all HAIs in the neurosurgical ICU, and 46% of the isolates were cultured from respiratory specimens [21]. More recent reports mention numerous cases of encephalitis caused by this microorganism [22]. In a multi-center study of ICUs with different profiles, *Acinetobacter baumannii* was found to be the predominant microorganism [23].

In a review, Montemurro et al. [24] emphasize the influence of the oral microbiota on some forms of infections in neurosurgery. Transient bacteremia caused by oral microbiota may lead to bacterial colonization in extraoral sites and, as a consequence, cause infections, e.g., brain abscesses. Awareness of this route of infection should be included in the surveillance of HAI and encourage staff to carry out a thorough physical examination in order to early detect and treat infections in the oral cavity, thus minimizing HAI.

Our study has some limitations. The study was based on a strict ECDC protocol comprising the definitions and criteria of HAIs and recommended active surveillance in ICU. However, as described above, a different approach was incorporated in both hospitals as regards the detection of infections. There was no validation of the process in the hospitals under study and no power analysis was performed when planning this study. ECDC protocol is predominantly dedicated to intensive care units and its implementation and subsequent data collection in the neurosurgery units are the strong points of our study.

## 5. Conclusions

The results of the surveillance of HAIs carried out in the two neurosurgical units studied point to several areas requiring intensification of activities aimed at reducing the number of HAIs and improving the safety of hospitalized patients. The data indicate the need for the daily infection control professionals work closely with patients in the units, which would be associated with early detection of PN, lower incidence rates of VAP and better detection of UTIs. Reliable HAI detection, analysis and feedback are essential elements in optimizing HAI prevention and control.

## Figures and Tables

**Table 1 ijerph-19-07544-t001:** Characteristics of the hospitals studied: neurosurgery departments, patients, use of invasive devices, incidence per 100 hospitalizations and 1000 person-days from 2014 to 2019.

Characteristics of/Hospital/Unit/Patients	Hospital A	Hospital B	Total
Hospital description			
Size (number of hospital beds)	620	530	1150
Infection Prevention Control Nurse (IPCN)–full-time employment	3	3	6
Hospital referral level	second	second	second
Characteristics of neurosurgical units			
Number of neurosurgical beds	35	28	63
Number of neurosurgical intensive surveillance beds	5	3	8
Number of patients	8347	3770	12,117
Number of surgeries	7799	3507	11,306
Number of patient-days (pds) of hospitalization	68,008	34,729	92,737
Invasive device utilization ratio			
Patient-days with urinary catheter	21,187	18,085	39,272
Urinary catheter UR *	0.31	0.4	0.36
Patient-days with central line catheter	5296	4000	9296
Central line catheter UR **	0.08	0.09	0.09
Patient-days with mechanical ventilation	580	379	959
Mechanical ventilation (MV) UR ***	0.01	0.01	0.01
Characteristics of patients			
Patient age: Me/IQR, *p* < 0.001	57/45–68	64/42–76	62/50–71
Sex F/M, *p* = 0.129	0.7	0.5	0.6
Days of stay: Me/IQR, *p* < 0.001	22/10–40	32/22–52	28/17–47
Healthcare-associated infections (HAI)			
Number of HAIs: OR (95% CI): 2.1 (1.73–2.58) *p* < 0.001	192	181	373
Incidence per 100 hospitalisations	2.3	4.8	3.1
Incidence density per 1000 pds	2.8	5.2	4.0

Person-days (pds) of hospitalization, standard deviation (SD), utilization ratio (UR), healthcare-associated infections (HAI), significance level (*p*), Odds Ratio (OR), 95% confidence interval (CI), median (ME), interquartile range (IQR), * patient-days/patient-days with urinary catheter, ** patient-days/patient-days with central vascular catheter, *** patient-days/patient-days with ventilation.

**Table 2 ijerph-19-07544-t002:** Clinical forms and numbers of HAIs, incidence of HAIs per 100 hospitalizations, Odds Ratio in the years 2014–2019.

HAI Type	Hospital A	Hospital B	Total	*p* Value	OR (95% CI)
Form of HAI *n* (morbidity %)					
Pneumonia (PN)	60 (0.7)	52 (1.4)	112 (0.9)	<0.001	1.918 (1.321–2.787)
Urinary tract infection (UTI)	21 (0.2)	87 (2.3)	108 (0.9)	<0.001	8.988 (5.686–14.792)
Surgical site infection (SSI)	77 (0.9)	19 (0.5)	96 (0.8)	<0.05	1.402 (0.788–2.494)
Bloodstream infection (BSI)	34 (0.4)	23 (0.6)	57 (0.5)	0.132	1.497 (0.881–2.546)
Total	192(2.3)	181 (4.8)	373 (3.1)	<0.001	2.142 (1.741–2.634)

Other types of HAI detected: 13 cases (incidence 0.1%) Gastrointestinal tract infection-Clostridioides difficile Infection (GI-CDI) and 9 cases (incidence 0.1%) Skin and soft tissue (SST), Healthcare associated infections (HAI), number (*n*), significance level (*p*), Odds Ratio (OR), 95% Confidence Intervals (95% CI).

**Table 3 ijerph-19-07544-t003:** Analysis of device-associated PN, UTI, BSI in neurosurgery units in the years 2014–2019.

HAIs Associated with the Use of Invasive Devices	Hospital A	Hospital B	Total
Ventilator-associated pneumonia (VAP)			
VAP	35	9	44
VAP incidence density per 1000 MV pds	60.3	23.7	45.9
Catheter-associated urinary tract infection (CA-UTI)			
CA-UTI	21	82	105
CA-UTI incidence density per 1000 UC pds	1.0	4.5	2.7
Central venous catheter-Bloodstream infection (CVC-BSI)			
CVC-BSI	7	11	18
CVC-BSI incidence density per 1000 CVCpds	1.3	2.8	1.9

Healthcare associated infections (HAI), Mechanical ventilation (MV), Urinary catheter (UC), Central venous catheter (CVC), Person-days (pds) of hospitalization.

**Table 4 ijerph-19-07544-t004:** Pathogens responsible for HAIs isolated in microbiological studies, and clinical forms of HAIs in neurosurgery in the years 2014–2019.

Pathogen Type	Clinical Forms of HAIs								
	BSI		PN		UTI		SSI		Total
	Hospital A	Hospital B	Hospital A	Hospital B	Hospital A	Hospital B	Hospital A	Hospital B	
	*n* (%)	*n* (%)	*n* (%)	*n* (%)	*n* (%)	*n* (%)	*n* (%)	*n* (%)	*n* (%)
Gram-positive cocci *n* (%)	14 (41.2)	14 (60.9)	8 (13.3)	4 (7.7)	2 (9.5)	16 (18.4)	41(53.2)	5 (26.3)	104 (28.0)
*Staphylococcus aureus*	5 (14.7)	7 (30.5)	6 (10.0)	1 (1.9)	1 (4.8)	0 (0.0)	20 (26.0)	3 (15.8)	43 (11.5)
*Staphylococcus epidermidis*	4 (11.8)	6 (26.1)	0 (0.0)	1 (1.9)	0 (0.0)	1 (1.1)	12 (15.6)	2 (10.5)	26 (7.0)
*Other Gram-positive*	5 (14.7)	1 (4.3)	2 (3.3)	2 (3.8)	1 (4.8)	15 (17.2)	9 (11.7)	0 (0.0)	35 (9.4)
Enterobacteriaceae *n* (%)	12 (35.3)	6 (26,1)	15 (25.0)	6 (11,5)	11 (52.4)	44 (50.6)	14 (18,2)	10 (52.7)	118 (31.6)
*Klebsiella pneumoniae*	3 (8.8)	2 (8.7)	6 (10.0)	2 (3.8)	4 (19.0)	18 (20.7)	5 (6.5)	5 (26.3)	45 (12.0)
*Escherichia coli*	3 (8.8)	1 (4.3)	3 (5.0)	2 (3.8)	6 (28.6)	23 (26.4)	2 (2.6)	2 (10.5)	42 (11.3)
*Enterobacter cloacae*	5 (14.7)	1 (4.3)	3 (5.0)	1 (1.9)	1 (4.8)	2 (2.3)	5(6.5)	0 (0.0)	18 (4.8)
*Other Enterobacteriaceae*	1 (2.9)	2 (8.7)	3 (5.0)	1 (1.9)	0 (0.0)	1 (1.1)	2 (2.6)	3 (15.8)	13 (3.5)
Non-fermenting Gram-negative bacteria *n* (%)	6 (17.6)	2 (8.7)	13 (21.7)	1 (1.9)	4 (19.0)	5 (5.7)	17 (22.1)	0 (0,0)	48 (12.9)
*Pseudomonas aeruginosa*	1 (2.9)	1 (4.3)	0 (0.0)	0 (0.0)	3 (14.3)	3 (3.4)	7 (9.1)	0 (0.0)	15 (4.0)
*Acinetobacter baumannii*	5 (14.7)	1 (4.3)	13 (21.7)	1 (1.9)	1 (4.8)	2 (2.3)	10 (13.0)	0 (0.0)	33 (8.8)
Other *n* (%)	2 (5.9)	0 (0.0)	1 (1.7)	0 (0.0)	4 (19.0)	3 (3.4)	0 (0.0)	0 (0.0)	10 (2.7)
*Candida spp.*	2 (5.9)	0 (0.0)	1 (1.7)	0 (0.0)	4 (19.0)	3 (3.4)	0 (0.0)	0 (0.0)	10 (2.7)
No microbiological confirmation *n* (%)	0 (0.0)	1 (4.3)	23 (38.3)	41 (78.9)	0 (0.0)	19 (21.8)	5 (6.5)	4 (21.0)	93 (24.9)
NO TEST	0 (0.0)	1 (4.3)	17 (28.3)	41 (78.9)	0 (0.0)	18 (20.7)	4 (5.2)	4 (21.0)	85 (22.8)
NO GROWTH *	0 (0.0)	0 (0.0)	6 (10.0)	0 (0.0)	0 (0.0)	1 (1.1)	1 (1.3)	0 (0.0)	8 (2.1)
Total *n* (%)	34 (100)	23 (100)	60 (100)	52 (100)	21 (100)	87 (100)	77 (100)	19 (100)	373 (100)

Healthcare-associated infections (HAI), *Clostridioides difficile* Infection (CDI)-13 cases; skin and soft tissue (9 cases), number (*n*), percentage (%), bloodstream infection (BSI), pneumonia (PN), surgical site infection (SSI), urinary tract infection (UTI), * no microorganisms were detected in the collected material.

## Data Availability

Data are available upon a reasonable inquiry from Marta Wałaszek.

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
