# Peer review of "Healthcare-Acquired Infection Surveillance in Neurosurgery Patients, Incidence and Microbiology, Five Years of Experience in Two Polish Units"

_ijerph, 2022, doi:10.3390/ijerph19127544_

Round 1
Reviewer 1 Report
The changes to the abstract make the manuscript more focused; however they raise two new issues.
You added "Both hospitals have participated 19 in the unified prospective system, based on continuous surveillance of HAIs (since 2012), imple- 20 mented in accordance with the protocol and definitions of the Healthcare-Associated Infections 21 Surveillance Network (HAI-Net) European Center for Disease Prevention and Control – protocol 22 version 4.3.
Which is exactly the same as methods "Both hospitals have participated in the unified prospective system, based on continuous 91 surveillance of HAIs (since 2012), implemented in accordance with the protocol and definitions of 92 the Healthcare-Associated Infections Surveillance Network (HAI-Net) European Center for Disease 93 Prevention and Control"
Which is verbatim restated in methods. Copying the text into the abstract identically is not ideal, please rephrase this.
You still present the following, which I have now disagreed with twice.
"The lower incidence of HAIs (2.3%) 212 recorded in hospital A was due to the lack of direct work with the unit"
Followed by the addition
"Unfortunately, on the basis of the obtained results, it cannot be concluded that the effective- 215 ness of HAI surveillance in the studied departments is similar - this hypothesis is difficult to prove 216 statistically (quantitatively)"
I cant accept this. You added that the relationship is difficult to test statistically, but made such a definitive statement previously. You need to remove this quote ""The lower incidence of HAIs (2.3%) 212 recorded in hospital A was due to the lack of direct work with the unit""
"The countries of Central and Eastern Europe, to which Poland belongs, are in the first stage of reform and are still building surveillance systems and creating an infection control infrastructure with European surveillance programs. Weak commitment, resource scarcity and a lack of expertise were identified. It is common to undercut official infection control statistics"
This is an interesting addition. I was not aware the Polish system had such problems; I am generally impressed by results I see from that nation.
Author Response
Comments and Suggestions for Authors
The changes to the abstract make the manuscript more focused; however they raise two new issues.
You added "Both hospitals have participated 19 in the unified prospective system, based on continuous surveillance of HAIs (since 2012), imple- 20 mented in accordance with the protocol and definitions of the Healthcare-Associated Infections 21 Surveillance Network (HAI-Net) European Center for Disease Prevention and Control – protocol 22 version 4.3.
Which is exactly the same as methods "Both hospitals have participated in the unified prospective system, based on continuous 91 surveillance of HAIs (since 2012), implemented in accordance with the protocol and definitions of 92 the Healthcare-Associated Infections Surveillance Network (HAI-Net) European Center for Disease 93 Prevention and Control"
Which is verbatim restated in methods. Copying the text into the abstract identically is not ideal, please rephrase this.
Answer: It was rephrased in abstract: Both hospitals ran (since 2012) the unified prospective system, based on continuous surveillance of HAIs designed and recommended by European Center for Disease Prevention and Control (protocol version 4.3) in Healthcare-Associated Infections Surveillance Network (HAI-Net).
You still present the following, which I have now disagreed with twice.
"The lower incidence of HAIs (2.3%) 212 recorded in hospital A was due to the lack of direct work with the unit"
Followed by the addition
"Unfortunately, on the basis of the obtained results, it cannot be concluded that the effective- 215 ness of HAI surveillance in the studied departments is similar - this hypothesis is difficult to prove 216 statistically (quantitatively)"
I cant accept this. You added that the relationship is difficult to test statistically, but made such a definitive statement previously. You need to remove this quote ""The lower incidence of HAIs (2.3%) 212 recorded in hospital A was due to the lack of direct work with the unit""
Answer: we stated that it is our assumption: Consequently, our assumption is that the lower incidence of HAIs (2.3%) recorded in hospital A was due to the lack of direct work with the unit.
"The countries of Central and Eastern Europe, to which Poland belongs, are in the first stage of reform and are still building surveillance systems and creating an infection control infrastructure with European surveillance programs. Weak commitment, resource scarcity and a lack of expertise were identified. It is common to undercut official infection control statistics"
This is an interesting addition. I was not aware the Polish system had such problems; I am generally impressed by results I see from that nation.
Reviewer 2 Report
Good.
Author Response
Comments and Suggestions for Authors
Good.
Answers: We would like to thank for acceptation
Reviewer 3 Report
We thank the authors for re-working this manuscript in which they describe the infectious disease surveillance process and results at two polish hospitals. Our main concern initially was that the goal of the manuscript was unclear and the authors have indeed done a significant job in making this clearer as well as downplaying the significance of the findings to just descriptive. Please remove all references of "daily hard work" in this manuscript. It is editorializing and non-scientific.
The conclusions in the abstract and conclusion section still need work: "The data indicate the need for daily hard work of infection control professionals in the units, which would be associated with early detection of PN and lower incidence rates of VAP and better detection of UTI.". Did the data the authors present show that there is earlier detection using infection control professionals? As compared to what? This is an over-reach conclusion. In the introduction the authors state: " The authors' previous experience indicates that the local infection control teams in Polish hospitals considerably marginalize the problem of non-SSIs and in units other than ICUs, hence this analysis especially takes into consideration non-SSIs in neurosurgery patients." Is there any data to back this up? Otherwise this is a subjective sentence with no relevance or support to this study.
There continues to be a need for extensive English grammatical corrections. Consider the title: Healthcare-acquired surveillance in neurosurgery patients, incidence and microbiology, 5 years of experience of two Polish units . It should state: Healthcare-acquired infection surveillance in neurosurgery patients, incidence and microbiology, 5 years of experience of two Polish. units
Author Response
Comments and Suggestions for Authors
We thank the authors for re-working this manuscript in which they describe the infectious disease surveillance process and results at two polish hospitals. Our main concern initially was that the goal of the manuscript was unclear and the authors have indeed done a significant job in making this clearer as well as downplaying the significance of the findings to just descriptive. Please remove all references of "daily hard work" in this manuscript. It is editorializing and non-scientific.
The conclusions in the abstract and conclusion section still need work: "The data indicate the need for daily hard work of infection control professionals in the units, which would be associated with early detection of PN and lower incidence rates of VAP and better detection of UTI.".
It was change and now is: Prospective surveillance ran by infection prevention and control nurse in the units is associated with better detection of infections expressed in morbidity, especially PN and UTI, and a lower risk of VAP.
Additional change for daily hard work: daily effective prospective – near to patients – work
Did the data the authors present show that there is earlier detection using infection control professionals? As compared to what? This is an over-reach conclusion. In the introduction the authors state: " The authors' previous experience indicates that the local infection control teams in Polish hospitals considerably marginalize the problem of non-SSIs and in units other than ICUs, hence this analysis especially takes into consideration non-SSIs in neurosurgery patients." Is there any data to back this up? Otherwise this is a subjective sentence with no relevance or support to this study.
Answer: It is hard to prove lack of data on HAI epidemiology in other settings or populatiopns. However, this remark perhaps will encourage us to prepare the review on this subject in future.
poprawiłam na: The authors' previous experience indicates that in Polish hospitals the problem of non-SSIs and in units other than ICUs are marginalize
There continues to be a need for extensive English grammatical corrections. Consider the title: Healthcare-acquired surveillance in neurosurgery patients, incidence and microbiology, 5 years of experience of two Polish units . It should state: Healthcare-acquired infection surveillance in neurosurgery patients, incidence and microbiology, 5 years of experience of two Polish units
Title is corrected.
Language correction was done and changes are marked in blue
Round 2
Reviewer 3 Report
We commend the authors on making the edits needed to make this manuscript, and the work they have done more clear for readers
This manuscript is a resubmission of an earlier submission. The following is a list of the peer review reports and author responses from that submission.
Round 1
Reviewer 1 Report
Title
After reading the manuscript, I think the title needs to be changed. I believe the true value of this manuscript is in assessing the differences resulting from different types of HAI control. The title does not do a good job of explaining this.
General comments
A large portion of this paper is explaining the differences between two hospitals in terms of infection control, which I think is truly important. However, the differences in the way infection control and surveillance is done need to be elucidated much more clearly in the methods.
This paper has some interesting data. It seems to me that it is torn between being a paper reporting on the epidemiology of HAI (which it does quite well) and a paper studying differences in infection control (which is not as well done but is very important. If the paper seeks to be the first, a descriptive study of HAI in neurosurgery, I would encourage you to focus more on this and make the differences in infection control an ancillary element of the paper (effectively, deemphasize this and mention it as a limitation). If the goal is to be the second, you need to clarify and quantify the differences in infection control between sites and explain why they are significant to science. Both have great potential to improve medical science; but I believe the current paper is trying to do too much and has become too confusing. Additionally, please review spelling of all organism names and recalculate all of table 4 (the numbers presented here are not possible).
I am truly surprised you did not find/report a single case of HAI meningitis/ventriculitis (HCAVM). This is one of my areas of research and we find it fairly frequently in neurosurgery patients. I would like to hear some explanation of why this was not discussed-were there truly no cases?
Detailed Comments
Abstract
“The greatest differences be- 26 tween departments were in the incidence of PN (p<0.001), UTI (p><0.001), SSI (p><0.05)”
Note that you have not explained what PN is at this point-I had to look it up later in the text. I recommend you spell out pneumonia and clarify the abbreviation. Please do provide the full description then the acronym for all such abbreviations.
I feel that you should explain some of the differences in infection control between the hospitals in the abstract. Half of your conclusions are about this comparison, so mention it more in the methods of the abstract.
Background
“Common clinical forms of HAIs among this patient population include 48 surgical site infection (SSI), pneumonia (PN), bloodstream infection (BSI) urinary tract in- 49 fection (UTI)”
Again I am surprised there is no mention of HCAVM here. It is common. I would add a reference or two to this phenomenon and discuss why you did not look for that.
“The problem of HAIs among neurosurgical patients will increase with 50 the implementation of increasingly complex techniques and methods to save the health 51 and lives of trauma or oncology patients, who predominate in this type of units, and with 52 the need for Intensive Care Unit (ICU) hospitalization and the growing problem of anti- 53 biotic resistance [11].”
This is very well said, though I would change this type of units to this type of unit/these types of units
“The aim of the study was to analyze the results of HAI surveillance conducted inde- 59 pendently in two neurosurgical wards in southern Poland and assess whether the results 60 expressed in epidemiological coefficients provide an unambiguous basis for assessing the 61 level of patient safety and in the broader context of ward epidemiology”
I do not like the wording of the main aim (epidemiological coefficient). This is a strange and counterintuitive term. Consider rewording here and in the abstract.
Table 1:
What do the asterisks (*) indicate? This should be described in a footnote
Methods
“The study was conducted between 2014 and 2019 in two neurosurgery units located 67 in southern Poland.”
Can we clarify the study period a bit more? Does this mean January 1 2014-December 31 2019?
“Statistical analysis was performed 105 using basic statistical parameters, i.e. mean, 95% Confidence Intervals (95% CI), standard 106 deviation (SD), significance level (p), Odds Ratio (OR). Pearson's chi-square independence 107 test, ANOVA test for quantitative variables were used to compare the frequency of vari- 108 ants of the qualitative trait.The level of significance was p <0.05”
This paragraph needs to be rewritten-I assume it was Pearson for binary and ANOVA for linear. I also do not follow what you did for qualitative here.
Results
“The application of mechanical ventilation was identical and stated 124 at 0.01”
- what? Needs clarification
“The most common HAI was pneumonia (n=112, incidence 0.9%) - the morbidity was 131 significantly higher in hospital B (p<0.001, OR=1,9,(1,3-2,8). Also urinary tract infection 132 (n=108, 0.9%) was significantly more often diagnosed in hospital B (p>< 0.001, OR=8.9, (5.7- 133 14.8). The epidemiology of surgical site infection (n=96, 0.8%), bloodstream infection 134 (n=57, 0.5%,), gastrointestinal system infection (n=13, 0.1%), skin and soft tissue (n=9, 135 0.1%) did not differ between wards A and B (Table 2).”
I applaud you for providing Cis with estimates. The numbers for some of these (GI, ST) are too low to support a realistic comparison.
“The CA-UTI and CVC-BSI rates were found higher for hospital B, while in case of 141 VAP incidence rates – the opposite was observed.”
Define CA-UTI and CVC-BSI. While I can look them up and figure out what they are, you need to spell these out then provide acronyms. I am also not certain there is value to subdividing to this level. 103/108 UTIS were CA, so why even report them as CA? I would say just report UTIs and be done with it. The paper has some great data but it is confusing; amalgamating these details would go a long way to improving the manuscript (and would not weaken the findings much).
“Also in case of SSIs there were differences in predominance, in hos- 161 pital A Staphylococcus aureus was the most frequently isolated (10 cases, 30.3%) while in 162 hospital B Klebsiella pneumoniae (5 cases, 26.3%), where Gram-positive granules and Enter- 163 obacteriacea were the most frequently isolated pathogens. GI-CDIs also occurred in both 164 hospitals and were induced by Clostridioides difficile (Table 4).”
What is Gram positive granule-this is not a morphology I am familiar with.
Discussion
“A higher incidence of HAI was observed in hospital B, which should not be associ- 182 ated with a high risk of infection, but rather with more effective detection of HAIs diag- 183 nosed also without microbiological confirmation - the high efficiency of the infection con- 184 trol team: frequent personal contact with ward staff, including detection of HAI at the 185 bedside and assessment of patients' health status based on clinical symptoms presented 186 by them in conjunction with microbiological and other test results, including imaging.”
This is an interesting explanation. I assumed, from viewing your data, that there was a higher risk of infection, but you make a good point that this could be a surveillance bias. However, you state this as fact when in reality you do not prove that the infection control team is any more efficient at site B vs site A. I believe you should present both the hypothesis that sites have differences in infection rates as well as the hypothesis that it is a surveillance bias-in the absence of true proof we cannot definitively state what is happening.
“The lower incidence of HAIs (2%) recorded in hospital 189 A was due to the lack of direct work with the ward.”
Again, you cannot prove this. Rephrase as an opinion.
“There were no frequent visits to assess 190 patient's clinical condition for early detection of possible HAIs”
This is the sort of difference that I would want to see quantified to allow better interpretation of the results.
“These concerns seem to be confirmed by the results obtained in the 221 2016-2019 HAI survey in ICUs of southern Poland, where it was shown that Klebsiella 222 pneumoniae and Acinetobacter baumannim were the most frequently isolated microorgan- 223 isms [22].”
Check the spelling of Acinetobacter
“The results of the surveillance of HAIs carried out in the two neurosurgical wards 233 studied show several shortcomings and indicate areas that require intensification of pro- 234 cedures aimed at reducing HAIs and improving the safety of hospitalized patients.”
I don’t agree with this statement. You have demonstrated that there is a difference in terms of bed occupancy, type of HAI, ant etiology. But you have not proven these are the results of shortcomings. The language here must be changed or present further data quantifying the differences in the way HAI control is done.
Table 2 (title)
“Clinical forms, number of HAIs, incidence of HAIs per 100 hospitalisations, Odda Ratio in 137 the years 2014-2019.”
Should be Odds ratio not Odda
Table 3
You use total on the far right side of the table, but for the incidence density figures, the numbers are not a sum. the total of 60.3 and 23.7 is not 45.9 (I think 45.9 is a weighted average). Consider renaming this.
I am not certain what the term “Razem” refers to-looks like it might be together?.
“Healthcare associated infections (HAI), Ventilator-associated pneumonia (VAP), Mechanical venti- 150 lation (MV), Catheter-associated urinary tract infection (CA-UTI), Central venous catheter - Blood- 151 stream infection (CVC-BSI), hospitalised person-days (pds)”
This footnote is very helpful for explaining your acronyms! It should be copied earlier in the text to help us understand.
Table 4
“Inne ziarenkowce”-I think this is other gram positive?
Many of these numbers do not sum properly. Consider the total column. 23.1 should be the total percent of Gram-positive cocci. This should then be equal to subheadings (Staph aureus, 10.1 + Staph epidermidis 5.5 + Others 9.2). But 10.1+5.5+9.2=24.8, not 23.1. Additionally, Total of gram positive cocci (23.1) + Enterobacteriaceae (30.7) + Non fermenting Gram neg (11.9)+ Other (3.1)+ non confirmed (27.3) should equal 100 but this sums to 96.1. Please either explain this more clearly or check the numbers in the table.
Reviewer 2 Report
The authors present an article highly interesting to the field of both public health and neurosurgery. Some aspects should be considered before publication:
-What are the abbreviations "T" and "BB" referring to?
-Did you actually ask for gender or do you refer to sex in biological terms? Please correct accordingly
-When presenting data as mean(average) and SD, such presentation infers normal distribution. Did you check for normality? If not please do so and present data as median/IQR if data are not normally distributed.
-Please add a paragraph discussing in detail the limitations and shortcomings associated with your article: some aspects were already mentioned (less testing often results in less frequent positive findings), but other aspects should be touched: how was the study designed? Did you prospectively analyse HAIs for 5 years? If so, did you perform an a priori power analysis? If not, please add information on the design and state "no power analysis was performed when planning this study"
-If you did not perform a power analysis, all null findings (e.g. Bloodstream infections) need to be interpreted with caution, as there may have been an observed difference, had your sample size been larger. Please add this aspect to other points of limitations
Reviewer 3 Report
Paper is well written, some points need to be improve:
- Lines 46-47 - " HAIs incidence rate in neurosurgery units varies from 2.1% [7] to 14.2% [6], and in neurosurgical intensive care it is as 47 high as 21.7% [8]. " Discuss better this point.
- Line 172 - "Our study showed that the incidence rate of HAIs was 2.9% (1.9% in hospital A and 5.0% in hospital B)" Incidence between the 2 hospitals is very different. Do the 2 hospitals have the same type of patients? or does one of the two accept more polytrauma and therefore patients with a lower GCS at the admission? If yes, this can be a limitation and it must be stated at the end of the paper.
- Lines 218-225. Some of these microorganisms can come from other site. Consider these refs: Multiple Brain Abscesses of Odontogenic Origin. May Oral Microbiota Affect Their Development? A Review of the Current Literature. Appl. Sci. 2021, 11, 3316. doi: 10.3390/app11083316%C2%A0%20 -- Acinetobacter baumannii. J Neurosurg. 2013 Dec;119(6):1656. doi: 10.3171/2010.7.JNS10577.
- The conclusion section is missing. what is the conclusion of this article? what did the authors prove new?
- What about antibiotics the use in neurosurgical patients? is there in those hospitals a preoperative protocols? report or discuss about it.
- Line 30 - "The daily hard work of the infection prevention and control nurse in the wards is associated with better detection of infections expressed in morbidity" Discuss about it in the discussion section.
Reviewer 4 Report
The authors look at HAI in neurosurgical patients in two hospitals in Poland over a 5 year period. Hospital B was smaller, with fewer patients/operations and these patients were older but had a shorter length of stay. Hospital B had higher rate of infections especially PNA and UTI. The microbiota is described. The authors conclude with "The data indicate the need for daily hard work of infection control professionals in the units, which would be associated with early detection of PN and lower incidence rates of VAP and better detection of UTI. Reliable HAI detection, analysis and feedback are essential elements in optimising HAI prevention and control."
Our most important feedback to the authors is that the goal of the manuscript is unclear. If the goal is to be able to state the concluding sentence as aobve, the manuscript falls far short of that. Essentially what has been achieved in this manuscript is a description of infection rates at two hospitals. "Daily hard work" would require that the authors describe in detail what is done now at each hospital. Compare that to another time period, say prior to 2012 and see what changes have occurred. To state "Reliable HAI detection, analysis and feedback are essential elements in optimising HAI prevention and control. To make this statement, the authors need to describe the accuracy of testing at these hospitals and what the process is.
In addition to the extensive re-working of the manuscript around one goal, we recommend closer attention to grammer. Some examples include "Other risk factors for HAIs are the use of invasive devices such as central venous catheters (CVC), urinary catheters (UC), AND mechanical ventilation (MV); bloodstream infection (BSI), AND urinary tract infection (UTI); continuous target surveillance of HAIs until 2012 (?) implemented according to the protocol; etc